# Spin-polarized self-trapped excitons in low-dimensional cesium copper halide

Ruiqin Huang[1,6], Longbo Yang[2,6], Feng Yang[1], Yuttapoom Puttisong [3], Qingsong Hu[4], Guixian Li[1], Jingnan Hu[1], Zhaobo Hu [5], Liang Li[1], Jiang Tang [2], Weimin Chen [3], Yibo Han [1] ✉, Jiajun Luo [2] ✉ & Feng Gao [3] ✉

Spin polarized excitons induced by spin injection from magnetic ion to a single quantum dot, has been considered as a basic unit of quantum information transfer between spin and photon for spin-photonic applications. However, this state-of-the-art technology has only been found with limited coupling strength and weak excitonic emission. Here, we demonstrate a spin-polarized self-trapped exciton naturally formed in the zero-dimensional lattice of cesium copper iodide. Upon excitation, the conversion from $Cu^+$ ion to spin-1/2 $Cu^{2+}$ ion results in an in-situ self-trapped exciton, which facilitates a local Jahn-Teller distortion and guarantees the strong spin-exciton coupling and near-unity excitonic emission efficiency. Consequently, a giant Zeeman splitting of −53 meV and an effective excitonic g-factor of −93.5 are observed from magneto-photoluminescence. More importantly, this nano-scale coupling can also be driven by an external electric field, which generates electro-luminescence with a circular polarization of 44.5% at 4.2 K and 8% at 300 K. The spin-optic properties of this copper compound will stimulate the fabrication of next-generation spin-photonic devices based on self-trapped excitons.

The integration of electrical, magnetic, and optical properties in a semiconductor nanostructure is pivotal for the advancement of spintronics[1–3]. For making a spin-photonic device, spin-polarized exciton was revealed in the diluted magnetic semiconductors (DMS) to facilitate the coupling between optoelectronic and spin properties[4–7]. The spatial overlapping of wavefunction between the magnetic ions and intrinsic exciton is crucial for constructing these coupling systems. This often necessitates the use of low-dimensional nanostructures such as $Mn^{2+}$ or $Cu^{2+}$ doped II−VI and IV−VI quantum dots (QDs), where the exciton was confined and the *sp-d* interaction was enhanced[8–10]. However, as shown in Fig. 1a, Wannier exciton exhibits a large Bohr radius with broad wavefunction (~5 nm)[11–13], significantly exceeding the spatial extent of the electronic wave function

of magnetic ions (~0.5 nm), resulting in limited coupling strength in DMS systems. Additionally, doping magnetic ions into semiconductors can lead to energy transfer from excitons to dopants[14,15], drastically reducing excitonic luminescence.

Unlike the weakly bound Wannier excitons, Frenkel excitons show tightly-binding excitonic wavefunction in atomic scale[16], enabling the effective coupling of magnetic ion and exciton. Self-trapped exciton (STE) is a unique Frenkel exciton with natural low-dimensional structurally confined holes[17], as shown in Fig. 1b, it is typically formed in soft lattice materials with strong electron-phonon coupling which induces local lattice distortions, and highly confines the exciton within nanometer range[18,19]. In the existence of a local magnetic ion, the atomic-scale spin-exciton coupling is expected to be greatly enhanced due to

[1]Wuhan National High Magnetic Field Center and School of Physics, Huazhong University of Science and Technology, Wuhan, China. [2]Wuhan National Laboratory for Optoelectronics (WNLO) and School of Optical and Electronic Information, Huazhong University of Science and Technology, Wuhan, China. [3]Department of Physics, Chemistry and Biology (IFM), Linköping University, Linköping, Sweden. [4]Hubei Key Laboratory of Low Dimensional Optoelectronic Materials and Devices, Hubei University of Arts and Science, Xiangyang, China. [5]School of Chemistry and Chemical Engineering, Jiangxi University of Science and Technology, Ganzhou, China. [6]These authors contributed equally: Ruiqin Huang, Longbo Yang. ✉e-mail: ybhan@hust.edu.cn; luojiajun@hust.edu.cn; feng.gao@liu.se

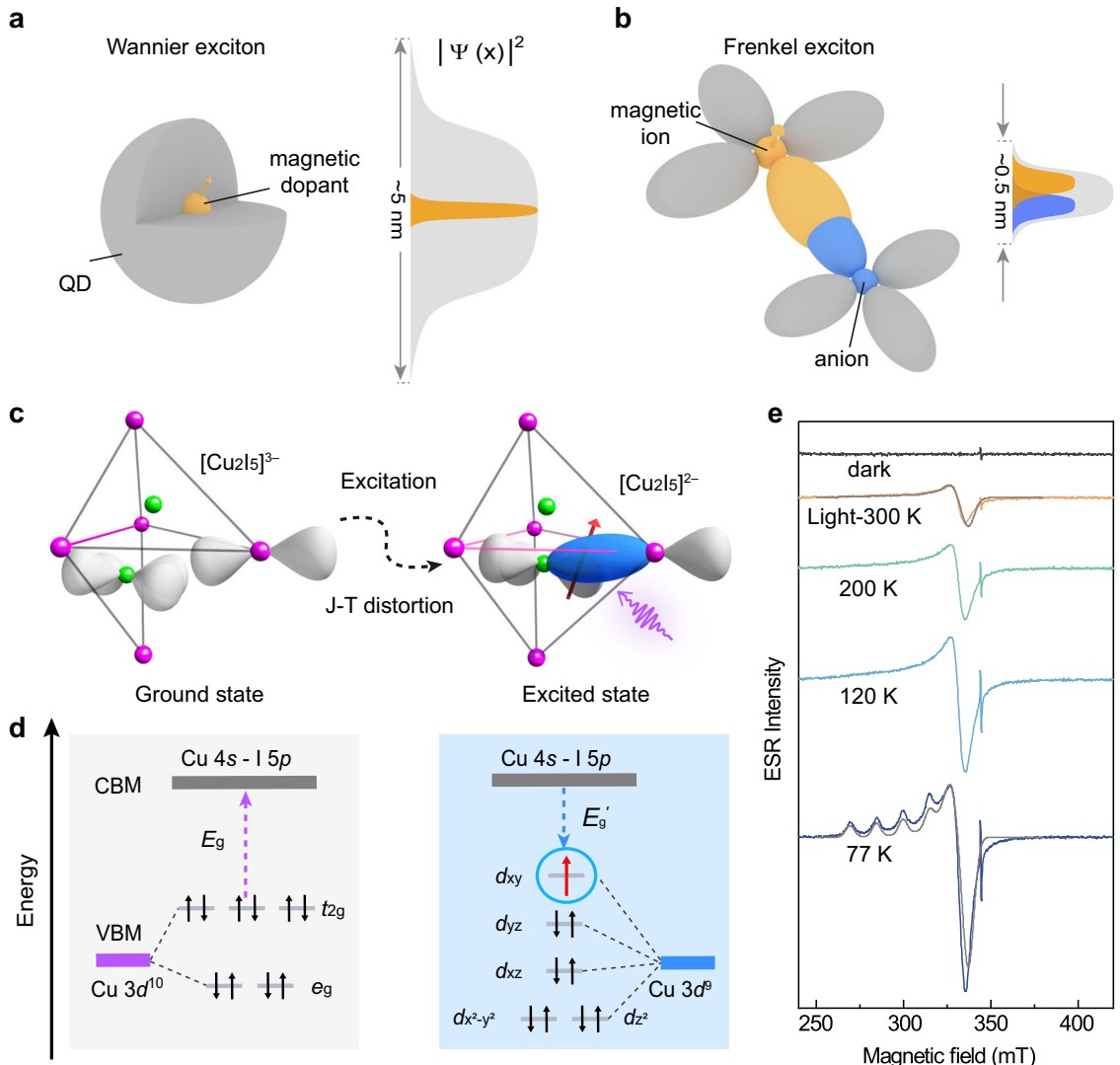

**Fig. 1 | The conversion of Cu⁺ to Cu²⁺ induces the Jahn-Teller effect upon photoexcitation of Cs₃Cu₂I₅. a** In conventional DMS QD, the overlapping of Wannier exciton wavefunction with that of magnetic dopant is small while for **b** STE (a unique Frenkel exciton), this overlapping is large. **c** The structural transition from [Cu₂I₅]³⁻ (left) to [Cu₂I₅]²⁻ (right) upon UV light excitation. The dumbbells represent the atomic orbital on the Cu and I sites; upon excitation, the I:5$p_z$ orbital and

Cu:3$d_{xy}$ orbital overlap (blue dumbbells) and drive the J-T distortion. **d** Schematic diagram of energy levels for ground and excited states of Cs₃Cu₂I₅ in tetrahedral crystal fields before and after UV excitation. **e** ESR spectra of Cs₃Cu₂I₅ without (dark) and with UV-light illumination (Light, $T$ = 300 K, 200 K, 120 K, 77 K), the gray curves show the simulation results considering a tetragonal J-T distortion.

the low dimension of this exciton[20]. Furthermore, if this magnetic center itself is part of the STE, the nonradiative energy transfer could be avoided and the excitonic emission would be enhanced. This intrinsic low dimensional exciton structure by the confinement effect of STE enhances the interaction between spin and exciton without the nanostructure fabrication, thus broadening their applicability. Therefore, spin-polarized STE deserves further investigation and in-depth research. Recently, a copper-based halide compound Cs₃Cu₂I₅ was found with an electronically zero-dimensional (0D) structure with typical STE emission[21–23], and can be a promising candidate for spin-photonics applications.

In this work, a strong local spin-STE interaction is revealed in Cs₃Cu₂I₅ from the observation of a giant Zeeman splitting and large circular polarization of luminescence in the presence of a magnetic field. As illustrated in Supplementary Fig. S1, it is found that the 0D structural component [Cu₂I₅]³⁻ transform to [Cu₂I₅]²⁻ upon excitation, charge conversion occurs, and the spinless Cu⁺ ions turn into spin-1/2

Cu²⁺ ions. The resultant structure is comparable to that of a DMS, but without energy transfer or surface effects. Leveraging the in-situ spin-STE interaction and the near-unity quantum yield of photoluminescence (PL), a prototype spin-photonic device has been fabricated, exhibiting the desired performance for circularly polarized light emission.

## Results and discussion

In light of this, Cs₃Cu₂I₅ thin films are prepared using single-source vacuum thermal evaporation. The homogeneous films prepared on quartz substrates exhibit orthorhombic structure at the growth temperature of 373 K (Supplementary Figs. S2–S4). PL peak at 445 nm with broad peak width (80 nm) originates from STE from the 0D [Cu₂I₅]²⁻ units (Supplementary Figs. S5–S7), exhibiting a PLQY of 95% (Supplementary Fig. S8). Cs₃Cu₂I₅ show large exciton activation energy and electron-phonon energy (Supplementary Fig. S9), with the PL lifetime of hundreds of nanoseconds (Supplementary Fig. S10). As shown in

Fig. 1c, d, the charge conversion from $[Cu^+]$–$3d^{10}$ to $[Cu^{2+}]$–$3d^9$ in tetrahedral coordination is stabilized by the Jahn-Teller (J-T) effect[21,24,25], which splits the valence states of $Cu^{2+}$ with an unpaired electron after excitation. This photo-assisted J-T effect instigates a conversion of the originally common line of the $CuI_3$ triangle and $CuI_4$ tetrahedron into two coplanar $CuI_4$ units. The transformation not only reduces the optical bandgap, but also induces an overlap between the $Cu:3d_{xy}$ orbitals and the $I:5p_z$ orbitals, providing strong coupling between localized $Cu^{2+}$ spin and the STE.

The above charge conversion and J-T effect can be evidenced by photon-excited electron spin resonance (ESR) spectroscopy (Supplementary Fig. S11). As shown in Fig. 1e, without illumination, there is no signal except a sharp transition at $g = 2.003$ (**B** = 350 mT), which is attributed to shallow halide vacancy defects formed during the sample growth process. In contrast, with the illumination of light at the wavelength of 280 nm, strong asymmetric ESR signals appear in the **B** range of 260−380 mT, revealing spin signals with strong magnetic anisotropy, which would be expected from a spin-1/2 electron at the $Cu^{2+}$ site under J-T distortion[26,27]. At cooling down from 300 K to 77 K, the thermal broadening of the ESR signal reduced, revealing a clear structure corresponding to electron-nuclear hyperfine interaction between electron spin $S = 1/2$ and nuclear spin $I = 3/2$ (corresponding with the Cu nuclear spin of the two natural abundance isotopes)[28]. From the fitting using an effective spin Hamiltonian analysis, the deduced anisotropic electron g-factors are $g_x \approx g_y = 2.07(\pm0.01)$, $g_z = 2.36(\pm0.01)$. The anisotropic hyperfine parameters are $A_x \approx A_y = 50(\pm20)$ MHz, $A_z = 490(\pm10)$ MHz. The symmetry evaluation of the $Cu^{2+}$ ESR signal suggests the symmetry of $C_{3V}$ or lower. A more detailed discussion can be found in Supplementary Note 1. The same parameters can be used to fit the ESR data at 300 K, by assuming the thermal broadening effect. We also note that under 450 nm illumination (below the absorption edge), no ESR signal was detected expect for the sharp resonance peaks attributed to intrinsic defects (Supplementary Fig. S12), which verifies the association of $Cu^{2+}$ to the above band edge absorption and the generation of STEs. In the X-ray photoelectron spectroscope (XPS) pattern, the $Cu$-$2p_{3/2}$ peak shows an energy shift towards the high energy side (Supplementary Figs. S13, S14), which indicates the electron loss upon UV light illumination. Additionally, the increased magnetic susceptibility under illumination also indicates the presence of magnetism in the excited state (Supplementary Fig. S15). These experimental results provide clear evidence of $Cu^+$ to $Cu^{2+}$ transition triggered by light illumination and the occurrence of tetrahedral-type J-T distortion in the local photo-excited $[Cu_2I_5]^{2-}$ structure components, which support previous theoretical conjectures[21].

The existence of photoinduced localized $Cu^{2+}$ in a locally J-T distortion lattice would result in the spin-exciton interaction, which then produces enhanced Zeeman energy splitting−energy splitting between the left-hand ($\sigma^+$) and right-hand ($\sigma^-$) circularly polarized PL peak energies denoted by $\Delta E = E^+ - E^-$. This $\Delta E$ is the energy splitting of STE, which is different from the former Zeeman splitting of the photo-excited $Cu^{2+}$ ions. As shown in Fig. 2a, the $\sigma^+$- and $\sigma^-$-polarized PL show opposite tendencies with increasing magnetic field in both intensity and peak energy, and both tendencies reverse when the field changes its polarities. The field-dependent $E^+$, $E^-$, and $\Delta E$ are shown in Fig. 2b, $\Delta E$ increases with **B** and saturates at $|\mathbf{B}| \geq 15$ T, with a saturation value of −53 meV. In the linear region at low fields ($|\mathbf{B}| \leq 10$ T), an effective exciton g-factor ($g_{eff}$) of −93.5 could be obtained. This giant Zeeman splitting for $Cs_3Cu_2I_5$ is much larger than that of Pb-based perovskites and comparable with that of some DMS nanostructures[8,29–31]. Further, $\Delta E$ and $g_{eff}$ for $Cs_3Cu_2I_5$ are negative, which is similar to the giant Zeeman splitting of Mn- or Cu-doped CdSe quantum dots (Table 1), where $sp$-$d$ exchange interaction occurs with energy level reversing and further splitting[9,32]. Therefore, we adopted the $sp$-$d$ exchange model for DMS to interpret the result for $Cs_3Cu_2I_5$, and the energy

splitting is demonstrated in the inset of Fig. 2b. The giant Zeeman splitting in $Cs_3Cu_2I_5$ should be due to the strong coupling between STE and localized $Cu^{2+}$ ion through the $3d_{xy}$-$5p_z$ orbitals, where the $Cu^{2+}$ itself is included for forming the band-edge excitonic states, rather than acting as a magnetic dopant. The $\Delta E$ can be fitted by the Brillouin function[8,9]:

$$\Delta E = g_{ex}\mu_B B + N_0(f_e\alpha - f_h\beta)\langle S_Z\rangle. \tag{1}$$

The first term represents the linear Zeeman splitting due to the intrinsic Zeeman effect of excitons, whereas the second term accounts for the additional splitting arising from the $s$-$pd$ exchange interaction. In this second term, $f_e$ and $f_h$ represent the exchange coupling strength between electrons and $Cu^{2+}$, as well as holes and $Cu^{2+}$, respectively, while $N_0\alpha$ and $N_0\beta$ denote the $s$-$d$, $p$-$d$ exchange interaction constants. The quantity $\langle S_Z\rangle$ is characterized by the Brillouin function which describes the magnetization of paramagnetic $Cu^{2+}$ moments, Further details are provided in Methods. An excellent fit using Eq. (1) was achieved with $N_0(f_e\alpha - f_h\beta) = 0.5$ eV. Additionally, the energy splitting decreases at rising temperatures according to the Boltzmann law, indicating the paramagnetic behavior of the excitonic spins[33], as shown in Fig. 2c, d. At $T = 30$ K, $\Delta E = -15$ meV fitted by Eq. (1) with the $sp$-$d$ exchange energy of 16 meV and $g_{eff} = -8.27$. At 60 K, $\Delta E = -10$ meV and $g_{eff} = -4.47$ (Supplementary Fig. S16).

This observation of giant Zeeman splitting and $g_{eff}$ in $Cs_3Cu_2I_5$ is similar to that in DMS. However, while the excitonic emission is suppressed in most DMS[8,9], in $Cs_3Cu_2I_5$, no nonradiative exciton-dopant energy transfer occurs. The superior performance of $Cs_3Cu_2I_5$ compared to typical DMS[34,35], demonstrated by the near-unity PLQY with remarkable $sp$-$d$ exchange interaction facilitate its application in spin-photonic devices (Fig. 2e). It is also found that the spin polarization of $Cs_3Cu_2I_5$ can be optimized by growth temperature, through which the grain size and crystallinity can be modulated. This variation alters the electron-phonon coupling strength, which in turn influences the formation of self-trapped excitons (STEs), and ultimately affects the degree of spin polarization (Supplementary Fig. S17−21). Accordingly, the $Cs_3Cu_2I_5$ thin film grown at 393 K is chosen for making a prototype of spin light-emitting diodes. A heterostructure of ITO/ZnO/$Al_2O_3$/$Cs_3Cu_2I_5$/MCP/TAPC/HAT-CN/Al is fabricated[36], utilizing the $Cs_3Cu_2I_5$ thin films as the active region (Supplementary Fig. S22). The working principle of spin-polarized electroluminescence (EL) is illustrated in Fig. 3a, where holes and electrons are injected and the two join in the $Cs_3Cu_2I_5$ layer to form STEs. According to the optical selection rules and conservation of angular momentum in the optical transitions[37], the excitonic recombination would produce circularly polarized luminescence, which can be attributed to the imbalanced population of electrons on the two pairs of Zeeman energy levels.

The magneto-EL shows similar behavior as that of magneto-PL. In Fig. 3b, the EL spectra show large circular polarization and Zeeman splitting at 4.2 K. The saturated $\Delta E$ is −20.4 meV at **B** ≥ 15 T with the $g_{eff}$ value of −27.2 (Fig. 3c). These values are in a similar magnitude to that in magneto-PL, and can be also attributed to the J-T distortion and accompanied $sp$-$d$ exchange interaction between the electric-field-driven $Cu^{2+}$ ions and STE. The circular polarization of EL, $P_{EL} = [(I^+ - I^-)/(I^+ + I^-)]$, where $I^+$ and $I^-$ represent the intensities of the $\sigma^+$- and $\sigma^-$-polarized photon emission, can be obtained from the EL spectra in Fig. 3d, and shown in Fig. 3e. At $T = 4.2$ K, the saturated $P_{EL}$ value reaches 44.5%, slightly larger than that of $P_{PL}$ (31%) (Supplementary Fig. S23). It is speculated that the reorganization of photon-generated excited states leads to spin dissipation[21], while the electron-hole pair directly recombines in the electric-field-driven excited state and show less spin dissipation. It is known that the excitonic spin polarization would be easily diminished by thermal fluctuation at rising $T$ for most semiconductor[29,38–40], however, even at room temperature, as demonstrated in Fig. 3d, e, the significant spin polarization of EL can

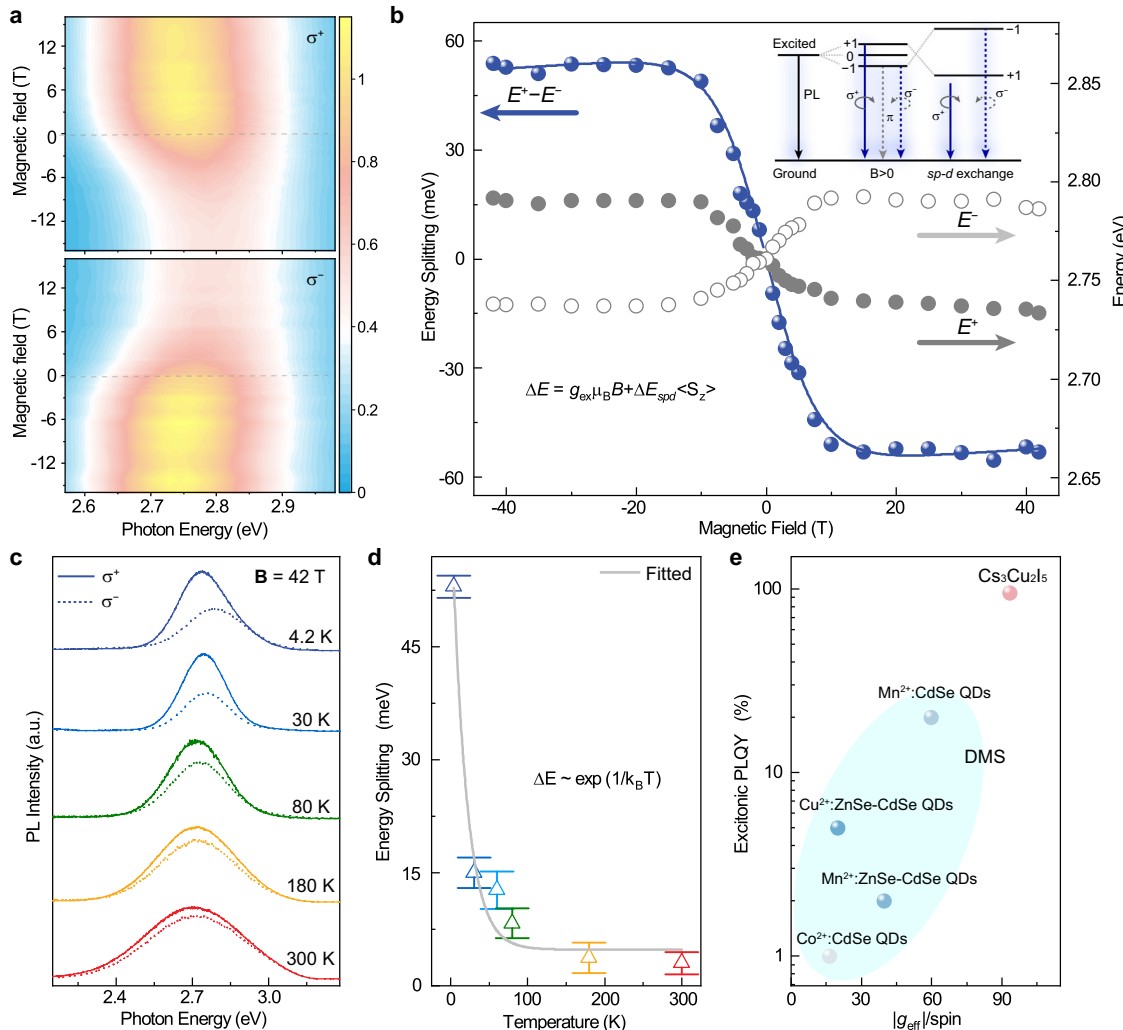

**Fig. 2 | Magnetic field effect on circularly polarized PL of the Cs₃Cu₂I₅ film. a** 3D pseudo-color plot of left- ($\sigma^+$) and right-circularly ($\sigma^-$) polarized PL spectra in magnetic fields from −16 T to 16 T at $T = 4.2$ K. **b** Peak energy of the $\sigma^+$ and $\sigma^-$ polarized PL and energy splitting ($\Delta E$) versus $B$. The inset illustrates the excitons energy levels, where the magneto-PL arises from the Zeeman splitting of $J_z = \pm1$ states. The reversal of the +1 and −1 states, due to the highly localized $Cu^{2+}$ ions, leads to the giant Zeeman splitting. **c** $\sigma^+$-and $\sigma^-$-polarized PL at different $T$. **d** $\Delta E$ versus $T$. **e** Comparison of exciton PLQY and effective exciton g-factor per spin in typical DMS materials ($Co^{2+}$:CdSe QDs[34], $Mn^{2+}$:ZnSe-CdSe QDs[8], $Cu^{2+}$:ZnSe-CdSe QDs[9], $Mn^{2+}$:CdSe QDs[35]) and Cs₃Cu₂I₅.

still be observed. The excitonic $g$ values were estimated using Brillouin function $P = \boldsymbol{P}_0 \tanh[g_{eff}\mu_B B/(2k_B T)]$, where $P_O$ is the saturation polarization. $g_{eff}$ values are 2.97, 3.11, and 3.06 at $T = 80$ K, 180 K, and 300 K, respectively, which are generally consistent with that of magneto-PL result shown in the Supplementary Fig. S23. Supplementary Fig. S24 shows the similar $T$-dependence of $P$ and $g_{ex}$ for both PL and EL at $B = 15$ T.

It is noteworthy that current-voltage curves of the heterostructure can be effectively modulated by the magnetic field, with an opposite tendency when **B** changes polarity at $T = 180$ K and 300 K. This indicates a significant magnetic field effect on the electric signals, driven by Zeeman effect-induced spin polarization in Cs₃Cu₂I₅, which leads to spin-selective transport through variations in scattering probabilities and band alignment (Supplementary Fig. S25). The above magneto-EL results reveal a new type of room temperature spin-photonic device based on the strong coupling between charge conversion-induced STE and magnetic ion at the same lattice site. It is important to note that the effective spatial confinement of STE around the $Cu^{2+}$ ion plays a crucial role in modulating the $sp$-$d$ interactions. Given this, it is reasonable to raise the question of whether this confinement can be adjusted by the dimension of the local lattice. In particular, variations in the local lattice dimension could potentially alter the spatial overlap between the

3$d$ orbitals of $Cu^{2+}$ and the 5$p$ orbitals of the surrounding halide ions, thereby influencing the strength of the $sp$-$d$ interactions and the overall spin polarization.

The extent of exciton confinement is comparatively studied in the 0D lattice of Cs₃Cu₂I₅ and the 1D lattice of CsCu₂I₃. As depicted in Fig. 4a, unlike the isolated Cu atoms in Cs₃Cu₂I₅, they are arranged along the crystal $c$-axis in CsCu₂I₃ (ref. 41). This higher dimension arrangement is expected to weaken the confinement of STE, thereby reducing the $sp$-$d$ exchange interaction and spin polarization. We successfully synthesized 1D CsCu₂I₃ using the antisolvent method and vacuum thermal evaporation (see Methods and Supplementary Figs. S26−S28). In Fig. 4b, c, we present the $T$-dependent PL spectra in the range of 80−300 K for the two thin films (see Supplementary Fig. S29 for PL spectra). The full width at half maximum (FWHM) of PL varies with $T$ in Fig. 4d, and fits using $FWHM = 2.36\sqrt{SE_p}\sqrt{coth[E_p/(2k_B T)]}$ results in $S = 95.5$ and 23.4, for Cs₃Cu₂I₅ and CsCu₂I₃, respectively, which indicates a more pronounced lattice distortion occurs in Cs₃Cu₂I₅ (ref. 42). The $P_{PL}$ values of CsCu₂I₃ at different $T$ are obtained by detecting circular polarized PL (Supplementary Fig. S30), and the exciton g-factors are derived from the linear fit of $P_{PL}$ versus **B** in Fig. 4e. A comparative analysis of $S$, $g_{ex}$, and $P_{PL}$ and for the distinct dimensional-confined samples is displayed in Fig. 4f,

**Table 1 | Magneto-optical properties of 3 $d$ ion-doped DMS and STE**

| Type | Material | T (K) | B (T) | ΔE (meV) | P_PL(%) | g_eff | Refs. |
|------|----------|-------|-------|----------|---------|-------|-------|
| DMS QDs | $Cu^{2+}$:ZnSe-CdSe | 1.7 | 7 | 4.3[a] | | +20[a] | 9 |
| | $Mn^{2+}$:ZnSe-CdSe | 1.6 | 6 | −14[a] | | −200[a] | 8 |
| | $Mn^{2+}$:CdSe | 1.8 | 5 | −54.6[a] | | −600[a] | 32 |
| | $Mn^{2+}$:(PEA)$_2$PbI$_4$ | 4 | 6 | | 13 | | 31 |
| | $Mn^{2+}$:CsPb(Cl/Br)$_3$ | 300 | 35 | | 4.6 | 1.25 | 14 |
| 0D STE | Cs$_3$Cu$_2$I$_5$ | 4.2 | 15 | −53 | 31 | −93.5 | This work |
| | | 300 | 42 | | 7 | 3.97 | |
| 1D STE | CsCu$_2$I$_3$ | 80 | 42 | | 13.9 | 0.8 | |
| | | 300 | 42 | | 5 | 1.07 | |

[a]Data are obtained from magnetic circular dichroism measurement.

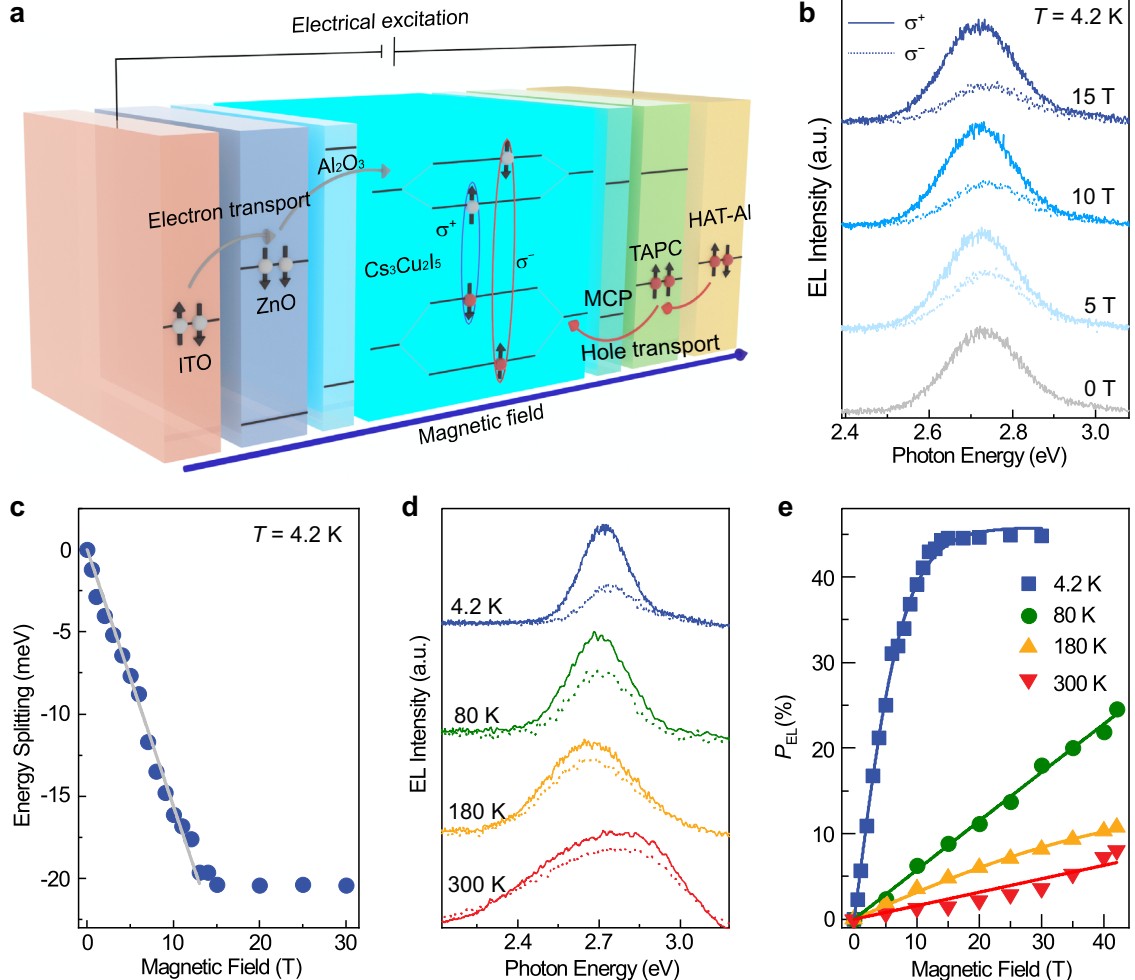

**Fig. 3 | Spin-polarized EL properties of the Cs$_3$Cu$_2$I$_5$. a** Schematic diagram of the transport and recombination of carriers with electrical excitation under the magnetic field. **b** Magnetic field-dependent σ$^+$- and σ$^-$- polarized EL spectra at $T$ = 4.2 K with an applied bias voltage of 16 V. **c** Energy splitting of EL as a function of **B. d** The σ$^+$- and σ$^-$- polarized EL spectra at $T$ = 4.2 K, 80 K, 180 K, and 300 K. **e** $P_{EL}$ versus **B**, with the solid curves fitted.

in which the 0D Cs$_3$Cu$_2$I$_5$ show superior spin polarization over the 1D CsCu$_2$I$_3$ due to a more pronounced quantum confinement effect and J-T effect.

In summary, we report the first observation of giant Zeeman splitting in an undoped compound, Cs$_3$Cu$_2$I$_5$, with high excitonic photoemission efficiency. In this zero-dimensional copper lattice, the Jahn-Teller distortion and charge conversion generate inherent local spin angular momentum which is conserved and transferred through the strong $sp$-$d$ exchange interaction from the optically or electrically Cu$^{2+}$

ions, to the self-trapped excitons in nano-scale range. Taking advantage of the spin-polarized self-trapped excitons, magnetic field-induced circular polarization of excitonic luminescence is achieved, and a prototype of a spin-photonic device is fabricated. Importantly, this series of compounds with condensed spin-1/2 copper exhibits significantly stronger spin-carrier interactions, brighter luminescence and greater flexibility in spin-electronic manipulation compared with DMS quantum dots. This study opens a new path for seeking optimal nano-materials for spin-photonic applications using self-trapped excitons.

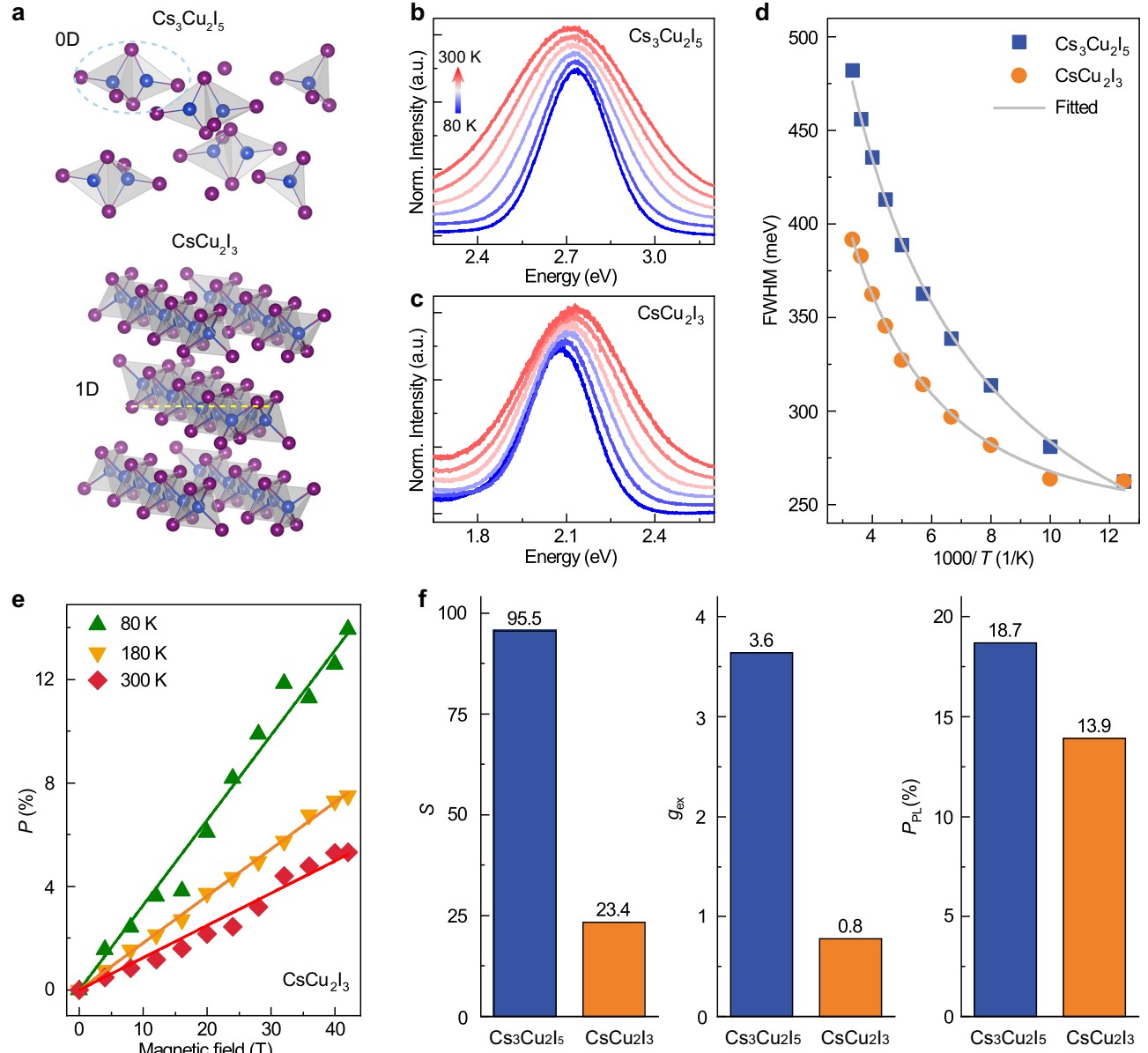

**Fig. 4 | Comparison of Jahn-Teller effect induced spin polarization between Cs$_3$Cu$_2$I$_5$ and CsCu$_2$I$_3$. a** The lattice arrangement of Cs$_3$Cu$_2$I$_5$ shows a 0D confinement structure of the Cu-I structure components, and that of CsCu$_2$I$_3$ exhibits a 1D structure. J-T distortion induced STE PL spectra at $T$ = 80-300 K for **b** Cs$_3$Cu$_2$I$_5$ and **c** CsCu$_2$I$_3$, respectively. **d** The fit of $T$-dependent FWHM of Cs$_3$Cu$_2$I$_5$ and CsCu$_2$I$_3$. **e** $P_{PL}$ versus **B** at different $T$ for CsCu$_2$I$_3$. **f** Comparison of $S$, $g_{ex}$, and maximal $P_{PL}$ of Cs$_3$Cu$_2$I$_5$ and CsCu$_2$I$_3$, respectively ($P_{PL}$ and $g_{ex}$ are obtained at $T$ = 80 K).

## Methods

### Preparation of Cs$_3$Cu$_2$I$_5$ powder

Cs$_3$Cu$_2$I$_5$ powder is synthesized by the anti-solvent method. 7.7943 g CsI (0.03 mol), 3.809 g CuI (0.02 mol), and 20 mL dimethyl sulfoxide (DMSO) are loaded into a centrifugal tube successively. The precursor in the centrifuge tube is shaken for 10 min for complete dissolution. The antisolvent dichloromethane (DCM, 60 ml) is added to the precursor solution. Then the precursor is centrifuged at 8170 $g$ to obtain the Cs$_3$Cu$_2$I$_5$ powder precipitation. Finally, the Cs$_3$Cu$_2$I$_5$ powder is dried and ball-milled for thermal evaporation.

### Preparation of Cs$_3$Cu$_2$I$_5$ film

Cs$_3$Cu$_2$I$_5$ film is prepared by single-source vacuum thermal evaporation (VTE). The Cs$_3$Cu$_2$I$_5$ powder is placed into the quartz crucible and heated to achieve a desirable deposition rate (-1 Å/s) under a high

vacuum (-10$^{-4}$ Pa). At the same time, the substrate is heated to 150 °C for in-situ annealing. It is worth noting that the material in the crucible needs to be completely evaporated since Cs$_3$Cu$_2$I$_5$ breaks down during evaporation (the thickness is determined by the mass of the material added to the crucible).

### Preparation of CsCu$_2$I$_3$ film

CsCu$_2$I$_3$ powder is synthesized by the anti-solvent method. 2.598 g CsI (0.01 mol), 3.809 g CuI (0.02 mol), and 20 mL DMSO are sequentially added to a centrifugal tube and shaken for 10 min to ensure complete dissolution. Subsequently, 60 ml of DCM is introduced as the anti-solvent, and the mixture is centrifuged at 8170 $g$ to obtain the CsCu$_2$I$_3$ powder precipitation. The obtained powder is then dried and ball-milled for thermal evaporation. CsCu$_2$I$_3$ film is prepared by single-source VTE, following the same experimental parameters as those employed in Cs$_3$Cu$_2$I$_5$ film preparation.

## Device fabrication

The pattern indium tin oxides (ITOs) are ultrasonic cleaned in detergent, ethanol, acetone, and deionized water successively for 30 min and dried by flowing nitrogen. The ZnO: PEIE precursors are spin-coated at 4000 rpm/s for 45 s, and annealed at 120 °C for 15 min. A 5 nm $Al_2O_3$ layer is deposited on ZnO: PEIE by atomic layer deposition (ALD). The ITO/ZnO: PEIE/$Al_2O_3$ substrates are then transferred into the vacuum chamber for the deposition of 100 nm $Cs_3Cu_2I_5$ emitting layer. After the substrates are cooled to room temperature, 10 nm MCP, 20 nm TAPC and 5 nm HAT-CN are sequentially evaporated on it at 0.2 Å/s rate. Finally, 60 nm Al is deposited as the anode. Noted that during all deposition processes, the substrates are rotated at 10 rpm/min under high vacuum (-$10^{-4}$ Pa). 1,1-Bis[4-[N,N'-di(p-tolyl)amino]phenyl]cyclohexane (TAPC), 2,3,6,7,10,11-Hexacyano-1,4,5,8,9,12-hexaazatriphenylene (HAT-CN), and 1,3-Bis(N-carbazolyl)benzene (MCP) were purchased from Xi'an Polymer Light Technology Co., Ltd.

## ESR and XPS measurements

ESR spectra were acquired in the temperature range of 77−300 K (A300-10/12, Bruker) at a microwave frequency of 9.651 GHz, with a microwave power of 20.11 mW. XPS measurements were performed using the Thermo Escalab 250Xi electron spectrometer with monochromatized Al Kα radiation (h$\nu$ = 1486.6 eV). Both investigations employed in-situ testing methodologies, with the excitation light wavelength of 280 nm.

## Magneto-PL/EL measurements

For PL measurements, multimode optical fibers are used to transmit excitation and emission beams. PL is excited by femtosecond lasers (840 nm, 76 MHz, 130 fs, Coherent) through a BBO crystal with triplet frequency generation to generate the 280 nm excitation and recorded by a spectrometer consisting of a monochromator (SP500i, Andor) and an electron multiplying charge-coupled device (EMCCD, Newton 970 P, Andor). The sample is positioned within a liquid helium cryostat, located at the center of a pulsed magnet capable of achieving a designed peak field strength of 60 T and a pulse duration of 400 ms. To ensure accurate measurements, the EMCCD is synchronized with the magnetic field pulses. For magneto-optical investigations involving circular polarization, left and right circular polarizers, comprising a quarter-wave plate and a thin film linear polarizer with axis angles of +45°/−45° respectively, are strategically positioned between the optical fiber and the sample detecting left and right circularly polarized PL. The difference between the EL of the device and the PL of the thin film is that the device's luminescence does not require excitation light to be excited but driven by a DC voltage (K2400 Keithley).

## Modified Brillouin function fit

Upon excitation, $Cu^{2+}$ ions are generated, leading to strong $sp$-$d$ exchange interactions with the 0D confined excitons. This interaction results in the observation of $\Delta E$ saturation at low temperatures and high magnetic fields. The average magnetization $\langle S_z \rangle$ can be characterized by the Brillouin function $S_z^{sat}B_S[g_{Cu}\mu_B H/(k_B T)]$, where the $S_z^{sat}$ represents the effective saturation value of $\langle S_z \rangle$ for each $Cu^{2+}$ ion, $g_{Cu}$ is the $Cu^{2+}$ g-factor, $\mu_B$ is the Bohr magneton, and $k_B$ is the Boltzmann constant. The Brillouin function is given by $B_S(x) = [(2S+1)/2S]\coth[((2S+1)/2S)x]-(1/2S)\coth[(1/2S)x]$, where for $Cu^{2+}$ ion, $S = 1/2$.

## Data availability

All data supporting the findings of this study are provided in the article and the Supplementary Information.

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

## Acknowledgements
Y.H. acknowledges the support from the National Key R&D Program of China (Grant No. 2022YFA1602702), and the Natural Science Foundation of China (Grant No. 11974126). J.L. thanks the support provided by the Natural Science Foundation of China (Grant No. 62322505). F.G. and W.C. acknowledge the support by Knut and Alice Wallenberg Foundation (Dnr. KAW 2019.0082) and the Swedish Government Strategic Research Area in Materials Science on Functional Materials at Linköping University (faculty grant SFO-Mat-LiU no. 2009-00971).

## Author contributions
Y.H., J.L., and F.G. conceived and supervised the experiments. R.H., L.Y., F.Y., and G.L. performed the optical measurements and analysed the data. Y.P. analysed ESR data. L.Y., Q.H., J.L., and R.H. prepared the sample, characterized the crystal structure and morphology of the samples. J.H. provided theoretical calculation of the sample structure. Z.H. performed photomagnetic measurements. Y.H. and R.H. wrote the paper in consultation with F.G., J.L., L.Y., J.T., L.L., W.C., and Y.P. all authors commented on the manuscript.

## Funding

## Competing interests
The authors declare no competing interests.
