## [Peer Review File · Nature Communications]

Spin-Polarized Self-Trapped Excitons in Low-Dimensional Cesium Copper Halides

Corresponding Author: Professor Feng Gao

Version 0:

Reviewer comments:

Reviewer #1

(Remarks to the Author)

Huang et al. have reported the spin-polarized self-trapped excitons in 0D Cs₂Cu₃I₅ system for next-generation spin photonic devices. I highly praise this manuscript's honest goal, efforts and work. This work is potentially important, and I recommend this report for publication after revision. I have attached my concerns and questions regarding this manuscript, which should be properly addressed.

1. In Figure 1(b), both in the title and description, there is a mistake in the spelling of "Frenkel" exciton.
2. In schematic diagram 1d, why the authors assigned the degenerate 3d orbitals as t_{2g} and e_g. Furthermore, here the authors have written I-5p orbital. However, in the figure caption and several places in the manuscript, it is written as I-6pz. Which one is correct?
3. In Figure S5, the authors have represented three spectra: PL spectra, PL excitation spectra, and UV-visible absorption spectra. For these spectra, the vertical-axis units are not the same.
4. In Figure S9(c), I do not understand the fitting logically. Some points are fitted, and the rest are not. Why? Fit it properly and then determine the values. Further, the authors did not explain each term of the equations, which is mandatory. Moreover, no citations from previous reports are available.
5. In Figure S10, please fit the PL decay curves and represent them. Further, the authors should mention the lifetime components with excited state contributions at different temperatures in a tabular format for better understanding.
6. In Figure S10, I am curious about the increase of the lifetime at lower temperatures (10K and 40K) compared to higher temperatures. What is the reason? Explain it and incorporate this in a separate section of the SI.
7. Without UV illumination, authors still show a sharp transition at 300K, and they have explained it due to an unknown grown-in defect. Without illumination, the structure only has diamagnetic Cu(I) present, which should be ESR silent. Please explain this. Also, I am very curious to know what an "unknown grown-in defect" is. How does it create a sharp transition in ESR?
8. In the experimental section, the authors claim that they have performed ESR and XPS measurements using in-situ testing methodologies with an excitation wavelength of 280 nm. It may be an interesting experimental setup for readers. Authors should carefully write all the details about the instrumentation with photographic images. From this, many researchers can understand the idea of instrumentation.
9. Authors should incorporate a separate note in the SI on the effective spin Hamiltonian analysis.
10. From the absorption spectra, it is clear that the material has no absorption at 450 nm. After 450 nm light excitation, authors claimed that they did not see any ESR signal, but I can see a strong peak with a g value of ~2.01 (approx.) Why? How does it verify the association of Cu²⁺ to the above band edge absorption?
11. the authors should provide original data and then show the fit line in HR-XPS spectra.
12. In Figure 3a, the authors show the upward movement of electrons and the downward movement of the holes. Please do the needful.
13. In the experimental section, the authors should write about the preparation methodology of CsCu₂I₃ compound with its PXRD and FESEM characterization.

Reviewer #2

(Remarks to the Author)

The authors of "Spin-Polarized Self-Trapped Excitons in Low-Dimensional Cesium Copper Halides" present results on the observation of the giant Zeeman effect, driven by the exchange coupling between excited Cu^{2+} ions and nearby self-trapped excitons (STE). Their experimental findings suggest that the photoresponse of $\text{Cs}_3\text{Cu}_2\text{I}_5$ qualitatively mimics the behavior of diluted magnetic semiconductors (DMS).

I found these results highly intriguing; however, in my opinion, the manuscript in its current form is not sufficiently mature and requires substantial revisions before being suitable for publication in Nature Communications.

My primary concern relates to the precise origin of the giant Zeeman effect. As the authors themselves note, the Zeeman splitting varies with growth conditions, suggesting an extrinsic origin of the magnetic moments interacting with STE. Yet, the discussion throughout the manuscript predominantly supports an intrinsic origin linked to the excitation of Cu cations to the +2 state. If the giant Zeeman effect indeed arises (only) from the exchange interaction between STE localized at CuI_4 units and excited Cu^{2+} cations, one would expect this interaction to remain consistently strong regardless of growth conditions. This inconsistency needs to be addressed in the revised version to ensure coherence of results and interpretation.

Also, the I-V characteristics are mentioned, but the description consists of only a few buzzwords, lacking any meaningful insight into the claimed magnetic field effect on the electrical signals or a more detailed explanation of the experimental observations:

Finally, certain parts of the manuscript are poorly written or difficult to understand. For example:

"However, as shown in Fig. 1a, in quantum information technology, Wannier exciton has..." – The reference to "quantum information technology" in this context is unclear.

"Exciton.. Self-trapped exciton (STE) is a unique Frenkel exciton with natural low-dimensional holes..." – The term "low-dimensional holes" is ambiguous. What exactly does this refer to?

"In this second term, f_e and f_h can be interpreted as the average percentage of electron and hole," – it is completely unclear what this percentage means.

"The copper-based halide compound $\text{Cs}_3\text{Cu}_2\text{I}_5$ shows a 0D structure..." – Is this referring to a spatially 0D structure or an electronically 0D structure?

Version 1:

Reviewer comments:

Reviewer #1

(Remarks to the Author)

The authors have addressed all the reviewers' queries and significantly enhanced the quality of the manuscript. The current form is recommended for publication in Nature Comm.

Reviewer #2

(Remarks to the Author)

The revised version of the manuscript exhibits significant improvement, and I appreciate that many of my initial concerns have been, at least partially, addressed. Nevertheless, a complete microscopic explanation for the influence of morphology on the observed Zeeman splitting is still not provided. That said, the direct correlation established between the Huang-Rhys factor and the magnitude of the Zeeman splitting stands as a particularly important finding. This leads me to wonder whether morphology-induced strain in the structure might play a role in modifying the phonon spectrum and, consequently, the exciton-phonon coupling.

Despite this lingering question, I consider the results presented in this article to be highly interesting. Even without a fully exhaustive microscopic understanding of every aspect, the work clearly merits publication. I hope it will serve as a catalyst for further research into the coupling between self-trapped excitons and the magnetic moments of the crystal lattice.

Reply to Reviewers' Comments

Reviewer #1 (Remarks to the Author):

Huang et al. have reported the spin-polarized self-trapped excitons in 0D Cs₂Cu₃I₅ system for next-generation spin photonic devices. I highly praise this manuscript's honest goal, efforts and work. This work is potentially important, and I recommend this report for publication after revision. I have attached my concerns and questions regarding this manuscript, which should be properly addressed.

Q1. In Figure 1(b), both in the title and description, there is a mistake in the spelling of "Frenkel" exciton

Reply: Thank you very much for pointing out the spelling error in Figure 1(b). We have carefully reviewed the manuscript and made the necessary corrections as follows:

Revised:

page 5, line 1: In Figure 1b, the label "Frankel" has been corrected to "Frenkel";

page 5, line 4: "a unique Frankel exciton" has been corrected to "a unique Frenkel exciton".

Q2. In schematic diagram 1d, why the authors assigned the degenerate 3d orbitals as t_{2g} and e_g. Furthermore, here the authors have written I-5p orbital. However, in the figure caption and several places in the manuscript, it is written as I-6pz. Which one is correct?

Reply: Thank you for the comment. In a coordination environment, the crystal field effect primarily governs the splitting of the 3d orbitals in Cu⁺/Cu²⁺. The underlying physical mechanism can be understood as follows: In the absence of an external crystal field, the five 3d orbitals (*d_{xy}*, *d_{yz}*, *d_{xz}*, *d_{x²-y²}*, *d_{z²}*) remain degenerate, possessing equal energy. However, when Cu⁺/Cu²⁺ is situated within a ligand field, the negatively charged ligands induce anisotropic electrostatic Coulomb repulsion, thereby lifting the orbital degeneracy and leading to an energy-level splitting. This splitting is governed by the interaction strength between the orbitals and the ligands, which is determined by the spatial orientation of the electron clouds relative to the ligand positions.

Specifically, in a tetrahedral coordination environment, the t₂ (*d_{xy}*, *d_{yz}*, *d_{xz}*) orbitals have electron clouds oriented toward the ligand positions, resulting in a stronger Coulomb repulsion and, consequently, a higher energy level. In contrast, the e (*d_{x²-y²}*, *d_{z²}*) orbitals have electron densities distributed along the tetrahedral faces rather than pointing directly toward the ligands, leading to weaker electrostatic repulsion and lower energy levels. Overall, the energy-level splitting of the 3d orbitals in a tetrahedral coordination field arises predominantly from electrostatic repulsion, which governs the potential energy landscape of the system.

In a tetrahedral coordination environment, the 3d orbitals of Cu²⁺(3d⁹) undergo crystal field-induced splitting, which can be further modified by the Jahn-Teller effect, leading to additional energy level splitting [J. Mater. Chem. C, 2023, 11, 2241–2250; Chem. Rev. 2013, 113, 1351–1390]. Specifically, the presence of a single unpaired electron in the t₂ orbital renders the Cu²⁺ system particularly susceptible to Jahn-Teller distortion. This distortion induces a structural deformation of the tetrahedral framework, leading to elongation or compression of ligands along specific directions alters the electrostatic interactions between the 3d orbitals and the surrounding ligands, thereby raising or lowering the energy of certain orbitals and ultimately lifting the original orbital degeneracy.

Furthermore, the electronic orbitals of the Γ^- ion are correctly identified as the I-5p orbitals. We regret the error and resulting confusion. The manuscript has been revised accordingly.

Revised:

page 5, line 6: “I:6p_z” has been corrected to “I:5p_z”;

page 6, line 8: “I:6p_z orbitals” has been corrected to “I:5p_z orbitals”;

page 9, line 14: “3d_{xy}-6p_z orbitals” has been corrected to “3d_{xy}-5p_z orbitals”.

Q3. In Figure S5, the authors have represented three spectra: PL spectra, PL excitation spectra, and UV-visible absorption spectra. For these spectra, the verticle-axis units are not the same.

Reply: Fig. S5 displays PL, PLE, and absorption spectra within a single graph, with the x-axis representing wavelength. Although the y-axis denotes different physical quantities, we have normalized the intensities of all three spectra to enable a direct comparison between the PL peak position and the absorption edge. This normalization highlights the Stokes shift and clearly reveals the spectral characteristics of STEs. We have provided additional clarification on this point in the Supplementary information related to Fig S5.

Revised:

Supplementary information (Fig S5), page 7, We have normalized the intensities of all three spectra to enable a direct comparison between the PL peak position and the absorption edge, revealing a Stokes shift of 162.2 nm, which is a characteristic feature of self-trapped exciton (STE) emission.

Figure S5. PL, PL excitation (PLE), and optical absorption spectra of the Cs₃Cu₂I₅ film, inset show the sample illuminated by white light and UV light, respectively.

Q4. In Figure S9(c), I do not understand the fitting logically. Some points are fitted, and the rest are not. Why? Fit it properly and then determine the values. Further, the authors did not explain each term of the equations, which is mandatory. Moreover, no citations from previous reports are available.

Reply: In Fig. S9, we employed the Arrhenius equation to fit the temperature-dependent PL intensity data. A noticeable decrease in PL intensity below 30 K, which we attribute to the enhanced formation of high-energy STE states at low temperatures, competing with the low-energy STE emission investigated in our study (Fig. R1). We hypothesize that multiple STE emission mechanisms exist in Cs₃Cu₂I₅, with the formation of both high-energy and low-energy STEs. The configuration coordinate diagram exhibits significant overlap in the high-energy region. At elevated temperatures, the high-energy STEs, characterized by a broad density of states, can either relax non-radiatively or

rapidly transition to the lower-energy state, resulting in the absence of detectable luminescence. In contrast, at low temperatures (below 30 K), the reduced phonon energy suppresses non-radiative recombination and relaxation pathways of high-energy STEs, favoring radiative recombination and giving rise to observable PL emission. We acknowledge the insufficient explanation of certain equations in the manuscript and greatly appreciate your valuable suggestion. To address this, we have included a comprehensive discussion of all fitting equations, along with relevant references, in the Supplementary Information.

Fig R1. (a) The configuration coordinate model of $\text{Cs}_3\text{Cu}_2\text{I}_5$ depicting the photophysical process above 30 K. (b) PL spectra of $\text{Cs}_3\text{Cu}_2\text{I}_5$ at 4.2 K and 300 K. The gray line represents the Gaussian fitting, while the cyan and purple solid lines correspond to the low-energy STE and high-energy STE emissions, respectively.

Revised:

Supplementary information (Fig S9), page 11:

Figure S9. (a) Temperature-dependent PL spectra for $\text{Cs}_3\text{Cu}_2\text{I}_5$ thin film. (b) PL spectra of $\text{Cs}_3\text{Cu}_2\text{I}_5$ thin film at 4.2 K and 300 K. The gray line represents the Gaussian fitting, while the cyan and purple solid lines correspond to the low-energy STE and high-energy STE emissions, respectively. (c) The configuration coordinate model of $\text{Cs}_3\text{Cu}_2\text{I}_5$ depicting the photophysical process above 30 K. (d) Temperature dependence of the full width at half-maximum (FWHM), The temperature dependence of the FWHM follows the relation [Materials Chemistry Frontiers, 2021, 5(19): 7088-7107]: $FWHM = 2.36\sqrt{S}E_p\sqrt{\coth[E_p/(2k_B T)]}$, where S is Huang-Rhys factors, E_p is the phonon energy, k_B is Boltzmann constant. By fitting the experimental data, the S is determined to be 68.49, and E_p is 11.51 meV. (e) PL intensity versus temperature. The temperature dependence of PL intensity can typically be described by the Arrhenius equation [Advanced Materials, 2018, 30(43)]: $I(T) = I_0/[1 + Ae^{-E_B/(k_B T)}]$, where I_0 represents the initial PL intensity (typically at 0 K), E_B is the activation energy of exciton thermal quenching. By fitting the experimental data, the exciton activation energy is determined to be 95.37 meV. Notably, at temperatures below 30 K, deviations from the fitting curve may arise from the emergence of high-energy STE emission. The suppressed phonon energy at low temperatures limits non-radiative recombination and relaxation pathways of these high-energy STEs, promoting radiative recombination and leading to detectable PL emission. This competing emission interacts with the primary low-energy STE, ultimately reducing its PL intensity.

Q5. In Figure S10, please fit the PL decay curves and represent them. Further, the authors should mention the lifetime components with excited state contributions at different temperatures in a

tabular format for better understanding.

Reply: Thanks for your valuable suggestion. We have now conducted a precise fitting of the PL decay curves and provided a detailed discussion in the Supplementary Information. To further enhance clarity, we have included a table summarizing the decay parameters of each component, facilitating a more comprehensive understanding for readers.

Revised:

Supplementary information (Fig S9), page 12:

Figure S10. Figure S10. (a) Time-resolved PL decay of the sample at different temperature, The black solid line represents the fitting curve (double exponential decay): $I = A_1 \exp(-t/\tau_1) + A_2 \exp(-t/\tau_2)$, where the τ_1 and τ_2 represent the fast and slow decay time, respectively, while A_1 and A_2 are the weighting factors of the corresponding decay components. (b) PL lifetime versus T . The average lifetime can be calculated using: $\tau_{ave} = (A_1 \tau_1^2 + A_2 \tau_2^2) / (A_1 \tau_1 + A_2 \tau_2)$. The table below summarizes the lifetime parameters obtained from biexponential decay fitting. Notably, a significant increase in lifetime is observed at 10 K, likely due to the suppression of non-radiative recombination as reduced phonon energy limits carrier relaxation pathways. Additionally, this effect may be influenced by the emergence of high-energy STE emission at low temperatures.

Q6. In Figure S10, I am curious about the increase of the lifetime at lower temperatures (10K and 40K) compared to higher temperatures. What is the reason? Explain it and incorporate this in a separate section of the SI.

Reply: At low temperatures, the emergence of a high-energy state induces energy transfer from the low-energy STE emission, leading to a decrease in PL intensity. However, time-resolved PL measurements reveal a pronounced increase in PL lifetime at 10 K, likely due to the suppression of non-radiative recombination as reduced phonon energy limits carrier relaxation pathways ([J. Phys. Chem. Lett. 2018, 9, 939–946], [Chem. Mater. 2020, 32, 3462–3468]). Additionally, this effect may be influenced by the emergence of high-energy STE emission at low temperatures.

Revised:

Supplementary Information (Figure S10), page 12, line 7:

“Notably, a significant increase in lifetime is observed at 10 K, likely due to the suppression of non-radiative recombination as reduced phonon energy limits carrier relaxation pathways ([J. Phys. Chem. Lett. 2018, 9, 939–946], [Chem. Mater. 2020, 32, 3462–3468]). Additionally, this effect may be influenced by the emergence of high-energy STE emission at low temperatures.”

Q7. Without UV illumination, authors still show a sharp transition at 300K, and they have explained it due to an unknown grown-in defect. Without illumination, the structure only has diamagnetic Cu(I) present, which should be ESR silent. Please explain this. Also, I am very curious to know what an "unknown grown-in defect" is. How does it create a sharp transition in ESR?

Reply: At 300 K, we observe a sharp ESR peak at $B = 350$ mT ($g = 2.003$). Notably, this feature appears regardless of light illumination, indicating that it reflects an intrinsic property of the material. We attribute this signal to shallow halide vacancy defects introduced during the sample growth process ([J. Am. Chem. Soc. 2018, 140, 15753–15763], [Adv. Mater. 2017, 29, 1700527]). The term “unknown intrinsic defects” lacks precision, and we have revised it accordingly in the manuscript. These iodine vacancies likely host isolated unpaired holes with a g -factor close to that of free electrons. Given their low concentration and weak spin interactions, a sharp and narrow signal is observed.

Revised:

page 6, line 12: “a sharp transition at $g = 2.003$ ($B = 350$ mT), which is attributed to shallow halide vacancy defects formed during the sample growth process.”

Q8. In the experimental section, the authors claim that they have performed ESR and XPS measurements using in-situ testing methodologies with an excitation wavelength of 280 nm. It may be an interesting experimental setup for readers. Authors should carefully write all the details about the instrumentation with photographic images. From this, many researchers can understand the idea of instrumentation.

Reply: In the Supplementary information, we have included photographs of the ESR and XPS setups, along with a detailed description of the experimental apparatus.

Revised:

Supplementary information (Fig S11), page 13:

Figure S11 Schematic diagram of the electron spin resonance (ESR) measurement setup. The photon-excited ESR system integrates a conventional ESR setup with a tunable light source to investigate spin dynamics under optical excitation. This system comprises a microwave source, resonant cavity, tunable magnet, light source (e.g., laser or LED), detector, and data acquisition unit. The experimental procedure includes: (i) Dark-state measurement: ESR signals of the sample are recorded in the absence of illumination to establish the baseline. (ii) Optical excitation: The sample is illuminated with a specific wavelength light source (280 nm) to excite charge carriers. (iii) Illuminated ESR measurement: Following 5 minutes of continuous illumination, ESR spectra are acquired in real time as a function of the magnetic field. (iv) Data analysis: The variation in the g-factor and the light-induced spin-state modulation are analyzed to elucidate the underlying light-spin interactions. The measurement was conducted using a Bruker, A300-10/12 spectrometer.

Supplementary information (Fig S13), page 15:

Figure S13 Schematic diagram of the X-ray photoelectron spectroscopy (XPS) measurement setup. The photon-excited XPS system incorporates a controlled light source into a conventional XPS setup, enabling direct investigation of surface electronic structures and chemical state evolution under illumination. The system comprises an X-ray source, an electron energy analyzer, an ultrahigh vacuum chamber, and a tunable light source—such as a monochromatic laser or LED—integrated via an optical delivery system. The experimental protocol involves three key steps: (i) dark-state measurements, where the $\text{Cs}_3\text{Cu}_2\text{I}_5$ sample is characterized in the absence of illumination; (ii) photoexcitation, in which the sample is irradiated with a 280 nm UV source for 30 min, followed by continuous exposure during XPS measurements; and (iii) data acquisition and analysis to resolve binding energy shifts, valence state transitions, and surface charge accumulation effects. The instrument model is Thermo Escalab 250Xi.

Q9. Authors should incorporate a separate note in the SI on the effective spin Hamiltonian analysis.

Reply: Thanks for your suggestion, we have added a detailed analysis of the effective spin Hamiltonian.

Revised:

page 7, line3: “A more detailed discussion can be found in Supplementary Note 1”.

Supplementary information page 32 (Supplementary Note1):

The spin Hamiltonian analysis is performed to determine possible microscopic structure and chemical origin of the defect responsible for the photo-active ESR signal. The effective spin Hamiltonian takes the form of $H = \mu_B \mathbf{B} \cdot \mathbf{g} \cdot \mathbf{S} + \mathbf{S} \cdot \mathbf{A} \cdot \mathbf{I}$, where μ_B is the Bohr magneton, \mathbf{B} is an external magnetic field, \mathbf{A} is hyperfine (hf) tensor. We note that though a ligand field and the

spin-orbit interaction are not explicitly included in the spin Hamiltonian, they are reflected by the g and A tensor anisotropy, that depend on orientation of ligand field and orbital character of the d -electron electronic states. This anisotropy is used to determine the local symmetry of the defect being probed by the ESR.

Q10. From the absorption spectra, it is clear that the material has no absorption at 450 nm. After 450 nm light excitation, authors claimed that they did not see any ESR signal, but I can see a strong peak with a g value of ~ 2.01 (approx.) Why? How does it verify the association of Cu^{2+} to the above band edge absorption?

Reply: Thank you for your correction. Indeed, $\text{Cs}_3\text{Cu}_2\text{I}_5$ does not exhibit absorption at 450 nm. Additionally, the description of ESR observations under 450 nm excitation in our manuscript requires clarification: experimental results indicate that upon 450 nm illumination, no additional ESR signals are detected beyond the pre-existing sharp absorption peak at $B = 350$ mT, which is already present in the dark state. We have revised and elaborated on this point in the manuscript. The ESR peak observed under these conditions should be attributed to the “unknown intrinsic defects” previously discussed.

By comparing ESR signals under 450 nm and 280 nm excitation, our study confirms that the formation of unpaired Cu^{2+} ions and the resulting ESR response occur only when the material absorbs excitation light (e.g., 280 nm), generating excitonic excitations accompanied by the Jahn-Teller effect. This finding reveals a direct correlation between Cu^{2+} active centers and the material's band-edge absorption.

Revised:

page 7, line 5: “We also note that under 450 nm illumination (below the absorption edge), no ESR signal was detected except for the sharp resonance peaks attributed to intrinsic defects (Supplementary Fig. S12)”

Q11. the authors should provide original data and then show the fit line in HR-XPS spectra.

Reply: Thank you for your suggestion. We have included a detailed fitting analysis of the original HR-XPS data in the Supplementary Information.

Revised:

Supplementary Information (Fig. S14), page 16:

Figure S14. (a) X-ray photoelectronic spectroscopy (XPS) pattern of the $\text{Cs}_3\text{Cu}_2\text{I}_5$ thin film with and without the illumination of 280 nm. High-resolution XPS element analysis shows (b) Cs-3d, (c) Cu-2p and (d) I-3d.

Q12. In Figure 3a, the authors show the upward movement of electrons and the downward movement of the holes. Please do the needful.

Reply: Thanks for your question. In Fig. 3a, we present a schematic illustration of the carrier recombination process in the LED heterostructure. The diagram is constructed based on the highest occupied molecular orbital (HOMO) and lowest unoccupied molecular orbital (LUMO) energy levels of each material layer, incorporating the energy distribution and transport mechanisms of electrons and holes. Given the inverted structure adopted in our LED design, under an applied electric field, electrons migrate along the descending energy levels, while holes move in the opposite direction, ultimately recombining within $\text{Cs}_3\text{Cu}_2\text{I}_5$, leading to light emission. This kind of illustration was widely used in LED structures ([Nat. Commun. 2024, 15, 6240], [ACS Energy Lett. 2021, 6, 4245–4254]).

Q13. In the experimental section, the authors should write about the preparation methodology of CsCu_2I_3 compound with its PXRD and FESEM characterization.

Reply: Thank you for your suggestion. We have now incorporated the preparation method of CsCu_2I_3 in the experimental section. Additionally, PXRD and FESEM data have been included in the Supplementary Information.

Revised:

page 17, line8: “**Preparation of CsCu_2I_3 film.** CsCu_2I_3 powder is synthesized by the anti-solvent method. 2.598 g CsI (0.01 mol), 3.809 g CuI (0.02 mol), and 20 mL DMSO are sequentially added to a centrifugal tube and shaken for 10 minutes to ensure complete dissolution. Subsequently, 60 ml

of DCM is introduced as the antisolvent, and the mixture is centrifuged at 7000 rpm/min to obtain the CsCu_2I_3 powder precipitation. The obtained powder is then dried and ball-milled for thermal evaporation. CsCu_2I_3 film is prepared by single-source VTE, following the same experimental parameters as those employed in $\text{Cs}_3\text{Cu}_2\text{I}_5$ film preparation.”

page 14, line 13: “We successfully synthesized 1D CsCu_2I_3 using the antisolvent method and vacuum thermal evaporation (see Methods and Supplementary Fig. S26-S28).”

Supplementary Information (Fig. S26), page 28:

Figure S26. XRD pattern of CsCu_2I_3 . The diffraction peaks at 10.77° , 13.46° , 21.64° , 26.19° , 27.11° , and 32.54° correspond to the (110), (020), (220), (102), (040), and (022) planes of the orthorhombic CsCu_2I_3 phase (JCPDS #45-0076), confirming the successful synthesis of orthorhombic CsCu_2I_3 with the Cmcm space group.

Supplementary Information (Fig. S27), page 29:

Figure S27. SEM image of CsCu_2I_3 . The film exhibits a uniform grain size distribution with few voids, indicating the synthesis of a high-quality CsCu_2I_3 thin film.

Supplementary Information (Fig. S28), page 30:

Figure S28. Elemental mapping and EDS spectrum of CsCu_2I_3 . The measured elemental composition of Cs, Cu, and I is 17.55%, 34.59%, and 47.86%, respectively, aligning well with the expected stoichiometry of CsCu_2I_3 . The elemental distribution maps further reveal a uniform spatial distribution of Cs, Cu, and I throughout the sample, confirming the successful synthesis of high-quality CsCu_2I_3 thin films.

Reviewer #2 (Remarks to the Author):

The authors of “Spin-Polarized Self-Trapped Excitons in Low-Dimensional Cesium Copper Halides” present results on the observation of the giant Zeeman effect, driven by the exchange coupling between excited Cu^{2+} ions and nearby self-trapped excitons (STE). Their experimental findings suggest that the photoresponse of $\text{Cs}_3\text{Cu}_2\text{I}_5$ qualitatively mimics the behavior of diluted magnetic semiconductors (DMS).

I found these results highly intriguing; however, in my opinion, the manuscript in its current form is not sufficiently mature and requires substantial revisions before being suitable for publication in Nature Communications.

Q1: My primary concern relates to the precise origin of the giant Zeeman effect. As the authors themselves note, the Zeeman splitting varies with growth conditions, suggesting an extrinsic origin of the magnetic moments interacting with STE. Yet, the discussion throughout the manuscript predominantly supports an intrinsic origin linked to the excitation of Cu cations to the +2 state. If the giant Zeeman effect indeed arises (only) from the exchange interaction between STE localized at CuI_4 units and excited Cu^{2+} cations, one would expect this interaction to remain consistently strong regardless of growth conditions. This inconsistency needs to be addressed in the revised version to ensure coherence of results and interpretation.

Reply: Thank you for your comment. We agree that the giant Zeeman effect is an intrinsic effect, and should not be changed by extrinsic origins. However, it is important to note that this giant Zeeman effect is caused by the exchange interaction between Cu^{2+} ions and STEs. Therefore, changes of the $3d(\text{Cu}) - 5p(\text{I})$ orbital overlap (confinement of STE) would modulate the exchange interaction; changes of STE itself (through exciton-phonon coupling) would also modulate the extent of Cu-STE interaction. These two factors modulate the circular polarization of photoluminescence (PL)—which manifests as the giant Zeeman effect—as follows:

1. Extent of Exciton Confinement: In Figure 4, the comparative study between 0D $\text{Cs}_3\text{Cu}_2\text{I}_5$ and 1D CsCu_2I_3 demonstrates that in strongly confined 0D materials, exciton-phonon coupling is significantly larger, thereby result in stronger Jahn-Teller distortion. Combined with the intrinsic electronic confinement of the 0D structure, this effect further strengthens the $sp-d$ exchange interaction between unpaired electron spins and excitons. Notably, this mechanism aligns closely with our experimental findings on the magneto-optical spectra of $\text{Cs}_3\text{Cu}_2\text{I}_5$ thin films with different grain sizes, further supporting the validity of our conclusions.

2. Grain Size of the Nanoparticles in Thin Films: The variation in the giant Zeeman splitting of $\text{Cs}_3\text{Cu}_2\text{I}_5$ thin films with growth conditions primarily arises from the influence of growth parameters on grain size distribution (Fig. S15, Fig. S19). The morphology and dimensions of the material play a crucial role in determining its physical properties. Our study reveals that $\text{Cs}_3\text{Cu}_2\text{I}_5$ films with different grain sizes exhibit distinct Huang-Rhys factors, indicating differences in electron-phonon coupling strength (Fig. S17). These differences affect the formation of STEs and modulate the Jahn-Teller effect, influencing Cu^{2+} formation and ultimately impacting the $sp-d$ exchange interaction between Cu^{2+} and excitons.

In summary, our results demonstrate a strong correlation between the giant Zeeman effect and the structural morphology of $\text{Cs}_3\text{Cu}_2\text{I}_5$. For the samples grown at different temperatures, as shown in Supporting Figure S11, the circular polarization and g-factor are almost proportional to the S-factor. This proportionality indicates that the giant Zeeman effect can be modulated by altering the

self-trapped exciton (STE) characteristics themselves. This finding suggests that optimizing the growth conditions of Cs₃Cu₂I₅ thin films offers an effective strategy for tailoring their spin polarization characteristics.

Revised:

page 10, line15: “This variation alters the electron-phonon coupling strength, which in turn influences the formation of self-trapped excitons (STEs), and ultimately affects the degree of spin polarization.”

page 13, line 3: “It is important to note that the effective spatial confinement of the self-trapped exciton (STE) around the Cu²⁺ ion plays a crucial role in modulating the *sp-d* interactions. Given this, it is reasonable to raise the question of whether this confinement can be adjusted by the dimension of the local lattice. In particular, variations in the local lattice dimension could potentially alter the spatial overlap between the 3d orbitals of Cu²⁺ and the 5p orbitals of the surrounding halide ions, thereby influencing the strength of the *sp-d* interactions and the overall spin polarization.”

Q2: Also, the I-V characteristics are mentioned, but the description consists of only a few buzzwords, lacking any meaningful insight into the claimed magnetic field effect on the electrical signals or a more detailed explanation of the experimental observations:

Reply: Thank you for highlighting this issue. We have provided additional discussions in both the manuscript and supplementary materials regarding the magnetic field effects on the electrical signals in Cs₃Cu₂I₅-based materials. The influence of the magnetic field on electrical transport arises from Zeeman effect-induced asymmetric spin distribution (spin polarization) in Cs₃Cu₂I₅. When the magnetic field direction is reversed (positive or negative field), the alignment of electron spin magnetic moments also flips (parallel or antiparallel to the field), leading to spin-selective transport due to variations in scattering probabilities and band alignment.

This spin-polarization control manifests as a linear shift and directional dependence in the I–V characteristics: under a positive magnetic field, parallel spin alignment enhances scattering, resulting in a high-resistance state and reduced current, whereas under a negative field, antiparallel spin alignment suppresses scattering, yielding a low-resistance state and increased current. Such magnetically induced resistance asymmetry reveals a strong coupling between spin degrees of freedom and charge transport in Cs₃Cu₂I₅, providing the fundamental physical basis for spintronic devices such as spin valves and memory units. By switching between high- and low-resistance states under positive and negative magnetic fields, binary information storage can be realized with low power consumption and high data density, overcoming the limitations of conventional charge-based storage technologies.

This mechanism not only confirms the pivotal role of spin polarization in macroscopic electrical responses but also provides both theoretical and experimental foundation for the design and optimization of magneto-electric coupling devices.

Revised:

page 12, line 20: “This indicates a significant magnetic field effect on the electric signals, driven by Zeeman effect-induced spin polarization in Cs₃Cu₂I₅, which leads to spin-selective transport through variations in scattering probabilities and band alignment.”

Supplementary information (Fig S25), page 27, line 5: “This spin-polarization control manifests as a linear shift and directional dependence in the I–V characteristics: under a positive magnetic field, parallel spin alignment enhances scattering, resulting in a high-resistance state and reduced current,

whereas under a negative field, antiparallel spin alignment suppresses scattering, yielding a low-resistance state and increased current. Such magnetically induced resistance asymmetry reveals a strong coupling between spin degrees of freedom and charge transport in $\text{Cs}_3\text{Cu}_2\text{I}_5$, providing the fundamental physical basis for spintronic devices such as spin valves and memory units.”

Q3: Finally, certain parts of the manuscript are poorly written or difficult to understand. For example: “However, as shown in Fig. 1a, in quantum information technology, Wannier exciton has...” – The reference to "quantum information technology" in this context is unclear.

“Exciton.. Self-trapped exciton (STE) is a unique Frenkel exciton with natural low-dimensional holes...” – The term "low-dimensional holes" is ambiguous. What exactly does this refer to?

“In this second term, f_e and f_h can be interpreted as the average percentage of electron and hole,” – it is completely unclear what this percentage means.

“The copper-based halide compound $\text{Cs}_3\text{Cu}_2\text{I}_5$ shows a 0D structure...” – Is this referring to a spatially 0D structure or an electronically 0D structure?

Reply: We appreciate these comments, which will help clarify terms that may cause confusion in the manuscript. We are committed to ensuring that our terminology is precise and easily understood, and carefully review and revise the relevant sections accordingly.

Q3-1: “However, as shown in Fig. 1a, in quantum information technology, Wannier exciton has...” – The reference to "quantum information technology" in this context is unclear.

Reply: We agree that the reference to “quantum information technology” in this context is unclear and not necessary. It has been removed in the revised manuscript to avoid confusion and to focus more directly on the specific topic being discussed.

Revised:

page 3, line 10: “However, as shown in Fig. 1a, Wannier exciton exhibit...”

Q3-2: “Exciton.. Self-trapped exciton (STE) is a unique Frenkel exciton with natural low-dimensional holes...” – The term "low-dimensional holes" is ambiguous. What exactly does this refer to?

Reply: Thank you for your comment. We agree that the term "low-dimensional holes" can be ambiguous. In the context of self-trapped excitons (STEs), these holes predominantly originate from hole localization within the low-dimensional structural confinement. This phenomenon constitutes a structure-induced dimensional constraint rather than a simple spatial limitation. For example, in $\text{Cs}_3\text{Cu}_2\text{I}_5$, the holes arise from zero-dimensional structural confinement, where strong electron-phonon coupling within isolated $[\text{Cu}_2\text{I}_5]^{3-}$ clusters leads to hole localization induced by Jahn-Teller distortion. To address this concern, we have revised the manuscript, we believe this revision provides a clearer and more precise description of the phenomenon.

Revised:

page 3, line 18: “we have updated the text to: "Self-trapped exciton (STE) is a unique Frenkel exciton with natural low-dimensional structurally confined holes...””

Q3-3: “In this second term, f_e and f_h can be interpreted as the average percentage of electron and hole,” – it is completely unclear what this percentage means.

Reply: We sincerely apologize for the confusion caused by the unclear expression. In the context of

our study, f_e and f_h represent the spatial overlap between the magnetic ions and the modulus-square of the wavefunctions Ψ_e and Ψ_h , respectively. Specifically, in the $\text{Cs}_3\text{Cu}_2\text{I}_5$ system studied in this manuscript, f_e and f_h correspond to the exchange coupling strength between electrons and Cu^{2+} , as well as holes and Cu^{2+} , respectively.

Revised:

page 9, line 20: “In this second term, f_e and f_h represent the exchange coupling strength between electrons and Cu^{2+} , as well as holes and Cu^{2+} , respectively.”

Q3-4: “The copper-based halide compound $\text{Cs}_3\text{Cu}_2\text{I}_5$ shows a 0D structure...” – Is this referring to a spatially 0D structure or an electronically 0D structure?

Reply: The term "0D structure" in our manuscript refers to an electronically zero-dimensional (0D) structure. This is characterized by the spatial confinement of charge carriers (electrons and holes) within isolated clusters.

Revised:

page 4, line 8: “The copper-based halide compound $\text{Cs}_3\text{Cu}_2\text{I}_5$ exhibits an electronically zero-dimensional (0D) structure with...”

Moreover, we have conducted a thorough review of the manuscript and meticulously refined specific phrasing to enhance clarity and precision.

Reply to Reviewers' Comments

Thanks for all the comments raised the reviewers, here we wrote our replies to each of them.

Reviewer #1 (Remarks to the Author):

The authors have addressed all the reviewers' queries and significantly enhanced the quality of the manuscript. The current form is recommended for publication in Nature Comm.

Reply: We sincerely appreciate your positive evaluation of our study, as well as your insightful comments, which have significantly contributed to the improvement of the manuscript.

Reviewer #2 (Remarks to the Author):

The revised version of the manuscript exhibits significant improvement, and I appreciate that many of my initial concerns have been, at least partially, addressed. Nevertheless, a complete microscopic explanation for the influence of morphology on the observed Zeeman splitting is still not provided. That said, the direct correlation established between the Huang-Rhys factor and the magnitude of the Zeeman splitting stands as a particularly important finding. This leads me to wonder whether morphology-induced strain in the structure might play a role in modifying the phonon spectrum and, consequently, the exciton-phonon coupling. Despite this lingering question, I consider the results presented in this article to be highly interesting. Even without a fully exhaustive microscopic understanding of every aspect, the work clearly merits publication. I hope it will serve as a catalyst for further research into the coupling between self-trapped excitons and the magnetic moments of the crystal lattice.

Reply: Thank you for your positive assessment of this work. We greatly appreciate your insightful comments. To further explore the relationship between the Huang-Rhys factor and Zeeman splitting, we are currently preparing additional samples with modulated nano-crystal sizes and local site symmetries through crystal engineering. We plan to submit the results of these experiments as an separate paper once new findings are achieved. We sincerely appreciate your encouraging feedback and hope that the current work will inspire further research in the field of spin-optics.